# An Image Encryption Algorithm Using Cascade Chaotic Map and S-Box

**DOI:** 10.3390/e24121827

**Published:** 2022-12-14

**Authors:** Jiming Zheng, Tianyu Bao

**Affiliations:** 1College of Computer Science and Technology, Chongqing University of Posts and Telecommunications, Chongqing 400065, China; 2Key Laboratory of Intelligent Analysis and Decision on Complex Systems, Chongqing University of Posts and Telecommunications, Chongqing 400065, China

**Keywords:** image encryption, cascaded chaotic system, S-box, diffusion

## Abstract

This paper proposed an image algorithm based on a cascaded chaotic system to improve the performance of the encryption algorithm. Firstly, this paper proposed an improved cascaded two-dimensional map 2D-Cosine-Logistic-Sine map (2D-CLSM). Cascade chaotic system offers good advantages in terms of key space, complexity and sensitivity to initial conditions. By using the control parameters and initial values associated with the plaintext, the system generates two chaotic sequences associated with the plaintext image. Then, an S-box construction method is proposed, and an encryption method is designed based on the S-box. Encryption is divided into bit-level encryption and pixel-level encryption, and a diffusion method was devised to improve security and efficiency in bit-level encryption. Performance analysis shows that the encryption algorithm has good security and is easily resistant to various attacks.

## 1. Introduction

After decades of development, digital information has grown extensively in capacity, and various electronic devices are changing gradually, accompanied by a large amount of data for transmission and communication [1], which brings various kinds of security risks in this background, especially image data and video. While traditional encryption techniques such as DES and AES have achieved good results in text encryption, they are not ideal for encrypting large amounts of modern image data [2,3]. 

Due to chaotic sensitivity to initial state and control value, good pseudo-randomness, ergodicity and unpredictable trajectory [4], the combination of chaos and encryption technology has produced many different chaos and applications: The chaotic system that has the property crosses a pre-define cylinder repeatedly and proposes the XOR approach for diffusion encryption of images. Despite receiving good encryption performance, this scheme lacks the appropriate scrambling operations, and the encryption scheme is independent of the plaintext image, making it difficult to resist chosen-plaintext attacks [5]. The encryption algorithm uses 3Dchua’s system with a combination of DWT transform and compressed sensing [6], and its experiments have shown its good encryption effect. The method NCCS, which generates new maps by combining methods, is able to overcome the shortcomings of a traditional one-dimensional chaotic map and gives a bit-level confusion and diffusion scheme while incorporating plaintexts in the keys used, which are shown to be well secured. However, when operating on the bit level, a large number of chaotic sequences are required, which creates some time consumption [7]. In addition, a cascaded chaotic system, as a form of chaos, is also a research hotspot; different chaos and encryption schemes are proposed: Zheng et al. [8] used an encryption scheme based on cascaded chaos, which generates new chaos through a cascaded one-dimensional chaotic map, so as to improve the performance of chaos and be used in encryption. Encryption methods based on DNA and dual chaotic systems are also proposed, but the overall encryption is not associated with the plaintext and lacks an efficient scrambling process. Lan et al. [9] proposed a composite integrated chaotic system, which integrated cascading, nonlinear combination and other operations, and then proposed an encryption scheme ICST with bit-level substitution and transform, but the overall lack of an effective diffusion process. Wang et al. [10] proposed a cascaded chaos model for CMCS to generate new chaos maps by using two-dimensional chaos mappings and one-dimensional chaos mappings, and demonstrated the good performance of the new chaos map through relevant experiments, an encryption method for chunking images and performing different operations on different blocks is also given, including bit-level transformations, shuffling algorithms, DNA encoding and V-shaped diffusion, etc. At the same time, in order to improve the security of the algorithm, people started to introduce some other techniques into chaotic encryption, and S-boxes is one of them.

Two types of cryptosystems can be classified: stream ciphers, which are converted bit-by-bit, and packet ciphers, which convert n-bit inputs into m-bit outputs. At the core of this conversion is the static S-box, which gives the cryptosystem the obfuscation properties described by Shannon [11]. An S-box is considered to be a well-performing S-box if it satisfies some of the following conditions [12]: bijection, nonlinearity, strict avalanche criterion (SAC) and output bit independence criterion (BIC). According to the way the S-box is generated, we can also classify it into static S-boxes and dynamic S-boxes. The security of the ciphertext is not guaranteed [13]. Dynamic S-boxes are based on key generation, and different keys can generate different S-boxes; all of these can improve the security of the encryption system [14]. Currently, S-boxes are the only nonlinear component of many packet ciphers, and the performance of S-boxes largely determines the security strength of encryption algorithms. In [15], Zhou et al. proposed a randomized approach to S-box generation by using DNA encoding and showed that the S-boxes generated by this method are resistant to different types of attacks. In addition, many researchers have applied knowledge from other fields to the generation of S-boxes, but some of these methods are still not efficient enough to meet today’s cryptographic efficiency requirements, so most of them cannot be practically applied in the encryption process. With the development of chaotic applications, many research results have been achieved in chaos-based S-box generation. Wang et al. [16] used LDCML to construct new S-boxes by dividing the interval of [0, 1] into 256 equal intervals, iterating LDCML to generate chaotic sequences, and generating non-repeating numbers between 0 and 255 according to the interval in which the generated values fall, and verifying that this method can generate S-boxes that satisfy the S-box criteria. Belazi et al. [17] used a map method to generate values between 0 and 255 and randomly place the mapped values through the sinusoidal map. In [18], Beg et al. finished S-box construction by expanding the remaining random values, generating chaotic sequences by iterating a chaotic map and expanding the remainder to continuously generate non-repeating values between 0 and 255 to add to the array.

As the security strength of the entire encryption algorithm is determined by the cryptographic strength of the S-box, the researcher has proposed many methods to enhance the S-box’s performance, such as using the high-dimensional chaotic system, and multiple chaotic systems to generate S-boxes have become solutions. Liu et al. [19] used a high-performance S-box construction using 3D chaotic systems and gave the corresponding encryption algorithm, iterating the high-dimensional chaotic systems to complete the S-box construction; however, the high-dimensional chaotic system then takes much time in computation, and although the security is improved, it becomes less efficient in terms of efficiency. Zheng et al. [20] proposed a multi-chaotic method for constructing dynamic S-boxes with improved efficiency and security and gave a corresponding encryption algorithm, which was shown to be feasible in experimental results. Özkaynak [21] proposed to use two S-boxes from the Henon map and Chen system, which are chosen at random to increase the encryption’s security. Wang et al. proposed to generate three S-boxes by the 3D chaotic map and perform one round of permutation for encryption [22], but the generation process generates a large number of useless chaotic sequences, which perform slightly worse in terms of real-time performance, and the key of the encryption algorithm is fixed, which means that the S-boxes generated by each encryption iteration are the same. Wang et al. proposed a cascade chaotic map and used it to generate S-box [23], followed by a diffusion operation after substituting with S-boxes, and this algorithm has good advantages in terms of security and complexity.

Based on the above analysis, a cascade chaotic map 2D-Cosine-Logistic-Sine map (2D-CLSM) is proposed in this paper. The initial values and parameters of the chaotic map are combined with the original image to resist known plaintext attacks. Then, an S-box construction method is proposed, and an S-box-based encryption method is designed. The encryption method is divided into four stages: key generation, S-box generation, bit-level encryption and pixel-level encryption. In bit-level encryption, a bit-level operation is performed on the plaintext pixel values, converting the original decimal plaintext image pixels to eight-bit binary, then permuting the lower four bits and using a proposed new diagonal diffusion method for the higher four bits. In the pixel-level encryption part, one diffusion is completed by a three-number XOR (by using the chaotic sequence value, the current pixel value to be encrypted and the previously encrypted pixel value), and the resulting value calculates the row and column index of the S-box for pixel value replacement. Through experimental analysis, the encryption algorithm in this paper has good security performance.

The remainder of this essay is structured as follows: Section 2 introduces the chaotic map; Section 3 describes the design of the encryption algorithm; Section 4 gives the simulation experiments and analysis of the results of the method titled in this paper; Finally, Section 5 gives conclusion remarks and further research work.

## 2. Introduction of Chaotic Map

In nonlinear dynamic systems, chaos is a stochastic process that is frequently employed in cryptography research [24]. One-dimensional Logistic and Sine map is classical chaotic map with a simple structure. They are defined as follows.
(1)$$t_{n+1}=4\mu t_{n}\left(1-t_{n}\right)$$
(2)$$t_{n+1}=ksin\left(\pi t_{n}\right)$$
with the control parameters μ,k∈[0,1].

The chaotic behavior of a chaotic system can be measured by the bifurcation diagram and the Lyapunov exponent. Figure 1a,b shows that the chaotic range of the one-dimensional Logistic chaotic map is restricted. From Figure 1b, we are able to find that μ is in the range of [0.89, 1], and its Lyapunov exponent is greater than 0. Figure 1c,d show that the chaotic range of the one-dimensional Sine map is similarly constrained, with k in Figure 1d in the range [0.87, 1] before its Lyapunov exponent is greater than 0 to be chaotic. Hence the classical Logistic and Sine maps are limited in terms of key space and are not resistant to brute force cracking.

### 2.1. Cascade Chaotic Map

A cascade map is a form of a chaotic map; the use of cascades can effectively improve the performance of chaotic systems. In order to solve the problem of small chaotic intervals and uneven distribution, we designed a two-dimensional cascade chaotic map, which has a more complex chaotic behavior than one-dimensional chaos and a faster iteration speed than two-dimensional chaos. The cascade system is shown in Figure 2, where $f_{1}\left(x_{n}\right),f_{2}\left(x_{n}\right)$ are two different subsystems.

The essence of cascading is that the output of a certain initial value after the iteration of system 1 is taken as the iterative output of system 2, and the iterative output of system 2 is taken as the iterative output of system 1, thus forming a circular iteration between two subsystems. Chaotic systems are generated by the cascade method, where the Lyapunov value of the system is the sum of the Lyapunov exponents of the cascaded subsystems, and the sequential trajectory of the system output deviates sharply as the number of iterations increases [25].

Using cosine functions is proposed and demonstrated in [26], where existing chaotic maps are cascaded by using cosine functions to improve their chaotic performance and extend the chaotic space, the expression as shown in Equation (3):(3)$$x_{i+1}=G\left(2^{h+F(x_{i})}\right)$$
where $G\left(\cdot \right)$ is the subsystem 2 and $F\left(\cdot \right)$ is the subsystem 1. In order to extend it to two dimensions, the general form expression is shown in Equation (4):(4)$$S\left(x_{i},y_{i}\right)=G\left(2^{h+F(x_{i},y_{i})}\right)$$
where $x_{i},y_{i}$ are two variables and G is the cosine that it can be represented as Equation (5):(5)$$\left\{\begin{array}{l} x_{i+1}=G\left(2^{h+A\left(x_{i},y_{i}\right)}\right)=cos\left(2^{h+A\left(x_{i},y_{i}\right)}\right)\\ y_{i+1}=G\left(2^{h+B\left(x_{i},y_{i}\right)}\right)=cos\left(2^{h+B\left(x_{i},y_{i}\right)}\right) \end{array}\right.$$

In Equation (3), to increase the chaotic complexity of the cosine function-based system, a new control parameter h is introduced as part of the exponential function. When the angle of the cosine function is large enough that even small differences can lead to large output differences such as $cos\left(2^{10}\right)=0.0001557636$ and $cos\left(2^{10.00001}\right)=0.987355203504$, also reducing the complexity of the calculation to make h∈[10,24].

$f_{1}\left(x_{i}\right)=4k_{1}x_{i}\left(1-x_{i}\right)$ and $f_{2}\left(x_{i}\right)=k_{2}\sin\left(\pi x_{i}\right)$ are Logistic map and Sine map, respectively, and $A\left(x_{i},y_{i}\right)=f_{1}\left(x_{i}\right)+f_{2}\left(y_{i}\right)$, $B\left(x_{i},y_{i}\right)=f_{1}\left(y_{i}\right)+f_{2}\left(x_{i}\right)$. We made them as subsystem 1 of Equation (4) and used it as the argument of subsystem 2, and then we derived Equation (6):(6)$$\left\{\begin{array}{l} x_{i+1}=cos\left(2^{h+f_{1}\left(x_{i}\right)+f_{2}\left(y_{i}\right)}\right)=cos\left(2^{h+\left(4k_{1}x_{i}\left(1-x_{i}\right)+k_{2}\sin\left(\pi y_{i}\right)\right)}\right)\\ y_{i+1}=cos\left(2^{h+f_{1}\left(y_{i}\right)+f_{2}\left(x_{i+1}\right)}\right)=cos\left(2^{h+\left(4k_{1}y_{i}\left(1-y_{i}\right)+k_{2}\sin\left(\pi x_{i+1}\right)\right)}\right) \end{array}\right.$$

Further, controlling the value of the parameter θ enables Logistic and Sine to be in chaos, which, according to Figure 1, makes $k_{1}=k_{2}=1-\frac{1}{9}\theta$ [27], and when using mod1 to control the iteration value between (0, 1), the chaotic map 2D-CLSM expression is shown in Equation (7):(7)$$\left\{\begin{array}{l} x_{i+1}=\left[cos\left(2^{h+(1-\frac{1}{9}\theta )\left(4x_{i}\left(1-x_{i}\right)+\sin\left(\pi y_{i}\right)\right)}\right)\right]\mathrm{mod}1\\ y_{i+1}=\left[cos\left(2^{h+(1-\frac{1}{9}\theta )\left(4y_{i}\left(1-y_{i}\right)+\sin\left(\pi x_{i+1}\right)\right)}\right)\right]\mathrm{mod}1 \end{array}\right.$$
where $x_{i}\in (0,1)$, $y_{i}\in (0,1)$ are the control parameters h∈[10,24], θ∈[0,1].

### 2.2. Performance Evaluation

In order to analyze the 2D-CLSM’s performance, we compared it with another existing 2D chaotic map for image encryption, i.e., the 2D Logistic-Sine-Coupling Map (2D-LSCM) [28].
(8)$$\left\{\begin{array}{l} x_{i+1}=\sin\left(\pi \left(\theta \cdot 4x_{i}\left(1-x_{i}\right)+(1-\theta )\sin\left(\pi y_{i}\right)\right)\right)\\ y_{i+1}=\sin\left(\pi \left(\theta \cdot 4y_{i}\left(1-y_{i}\right)+(1-\theta )\sin\left(\pi x_{i+1}\right)\right)\right) \end{array}\right.$$

#### 2.2.1. Chaotic Trajectory

The trajectory of a 2D-CLSM shows how motion increases over time from a specific initial state. In the case of periodic motion, the trajectory would be a closed curve, whereas the trajectory of chaotic behavior would theoretically never close or repeat. Therefore, chaotic trajectories usually occupy a part of the phase space and can reflect the randomness of the chaotic system output. If a chaotic trajectory can occupy a larger portion of the phase space, the chaotic system has a better stochastic output.

From Figure 3a–f, we can see that the 2D-CLSM is able to occupy the full phase plane for all trajectories within the parameter range, and the ability to occupy the full phase plane with both different θ and h indicates that the improved chaotic system has a better random output. On the contrary, 2D-LSCM is influenced by control parameters, which are not able to occupy full space.

#### 2.2.2. Bifurcation Diagram

We set initial value as $x_{0}=0.4,y_{0}=0.3$ and the 2D-CLSM h=15. As shown in Figure 4a–d, 2D-CLSM is in a chaotic state in the whole parameter domain, and it is more uniform. On the contrary, the 2D-LSCM does not occupy the entire plane, where the control parameters are at [0.3, 0.43], and values between [0, 0.1] cannot be generated iteratively.

#### 2.2.3. Lyapunov Exponent

The LE (Lyapunov exponent) describes the sensitivity of a chaotic mapping to initial values. In general, a chaotic map is in a chaotic state when λ>0 indicates that two adjacent phase points are about to separate and the chaotic map is in a chaotic state. For a two-dimensional map, the system of difference equations is assumed to be:(9)$$\left\{\begin{array}{l} x_{i+1}=f_{1}\left(x_{i},y_{i}\right)\\ y_{i+1}=f_{2}\left(x_{i},y_{i}\right) \end{array}\right.$$

Its Jacobian matrix at the point $x^{\left(i\right)}=$$\left(x_{i},y_{i}\right)$ is as follows:(10)$$f^{\prime }\left(x^{\left(i\right)}\right)={\left.\left[\begin{array}{cc} \frac{\partial f_{1}}{\partial x} & \frac{\partial f_{1}}{\partial y}\\ \frac{\partial f_{2}}{\partial x} & \frac{\partial f_{2}}{\partial y} \end{array}\right]\right|}_{\left(x_{i},y_{i}\right)}$$

Let $J_{i}=f^{\prime }\left(x^{\left(0\right)}\right)f^{\prime }\left(x^{\left(1\right)}\right)\cdots f^{\prime }\left(x^{\left(i-1\right)}\right)$, then the eigenvalue of $J_{i}$ may be expressed as $\lambda_{1}^{\left(i\right)},\lambda_{2}^{\left(i\right)}$. The Lyapunov exponents of system (9) can be expressed as Equation (11):(11)$$\lambda_{k}=\underset{i\to \infty }{\lim}\frac{1}{i}ln\left|\lambda_{k}^{\left(i\right)}\right|$$

The larger the value of λ, the faster the separation of point phase points in the phase space and the greater the sensitivity of chaos to initial values. Figure 5 shows the Lyapunov exponent of 2D-CLSM and 2D-LSCM; it is clear that the 2D-CLSM has a better chaotic behavior than 2D-LSCM.

#### 2.2.4. NIST Test

In this paper, 15 NIST tests were used to test the randomness of the generated sequences, and 15 correspond to *p*-values to show the test result. The sequence is considered random when the *p*-value is over 0.01. Table 1 contains the NIST test results for the 2D-LSCM map and the 2D-CLSM map. The NIST test results for the 2D-CLSM map are all “Success”, as shown in Table 1, and 11 of the 2D-CLSM test results are higher than 2D-LSCM, demonstrating 2D-CLSM map can generate pseudo-random sequences with good random performance.

#### 2.2.5. Information Entropy

In information theory, information entropy is used to quantify the uncertainty of the information content. It can be used to evaluate how random a set of data is. We transformed the chaotic sequence obtained by iteration into values between 0 and 255 and obtain the information entropy according to Equation (12):(12)$$H\left(X\right)=-\sum_{i=0}^{2^{N-1}}Pr\left(x_{i}\right)\log_{2}Pr\left(x_{i}\right)$$
where X is a data sequence, $x_{i}$ is the ith possible value in X and $Pr\left(x_{i}\right)$ is the probability of $x_{i}$. A bigger information entropy value means better randomness; for a set with 256 states, its maximum expected value is $\log_{2}{}^{256}=8$.

Better randomness is correlated with larger information entropy values, and Figure 6 shows the information entropy of the output sequences generated by 2D-CLSM for different parameter settings. From Figure 6, the mean information entropy value of 2D-CLSM is bigger than 2D-LSCM, and it is close to 8, which means good randomness.

## 3. New Encryption Algorithm Design

There are four main stages in our encryption algorithm: key generation, S-box generation, bit-level encryption and pixel-level encryption. The size of the plaintext image P is M×N, and t,μ,θ,x,y,Δm are the initial keys. The overall process is shown in Figure 7.

Two different chaotic maps are used in our algorithm: Logistic and 2D-CLSM, which are used in different stages of encryption. In the S-box generation stage, a Logistic map is used to generate the chaotic sequence needed to generate the S-box; a 2D-CLSM map is used in the bit-level operation and pixel-level operation phases. By using the different chaotic maps in different stages to expand the key space, a more complex encryption algorithm is generated.

### 3.1. Keys Generation

If the key stream used for encryption is only related to the key and not related to the plaintext image, the designed algorithm is not safe for chosen/known plaintext attack. In order to ensure the security of encryption, the encryption key is generated from the initial key and the plaintext image.

Initial encryption keys contain six decimal numbers, t,μ,θ,x,y,Δm. Six decimal encryption keys, $t_{0},\hat{\mu },\hat{\theta },x_{0},y_{0},\hat{h}$, are generated by combining the plaintext images, which are used as initial values and control parameters of the chaotic system. Using Δm>0 as a scrambling value prevents an attacker from attacking the key with a black or white image. The encryption key is generated by Algorithm 1, where $t_{0},\hat{\mu }$ is used as the initial value and control parameter of the Logistic map; $x_{0},y_{0},\hat{h},\hat{\theta }$ is set as the initial value, control parameter and variables h of 2D-CLSM.
**Algorithm 1** Generation of the encryption keysInput: Plain image P, initial keys t,μ,θ,x,y,Δm
Output: The encryption keys
$t_{0},\hat{\mu },\hat{\theta },x_{0},y_{0},\hat{h}$
   1: Read the size:
[M,N] = size (P)
   2: Obtain the sum of all pixels and add scramble number 
    $sum=\sum_{i=0}^{M-1}\sum_{j=0}^{N-1}P\left(i,j\right)+\Delta m$
   3: Calculate the mean of the all pixels
$\bar{m}=sum/\left(M\times N\right)$
   4:
$t_{0}=cos\left(\bar{m}\times 2^{t+15}\right)\mathrm{mod}1$
   5:
$\hat{\mu }=1-cos\left(\bar{m}\times 2^{\mu +15}\right)\mathrm{mod}0.1$
   6:
$\hat{\theta }=cos\left(\bar{m}\times 2^{\theta +20}\right)\mathrm{mod}1$
   7:
$x_{0}=cos\left(\bar{m}\times 2^{x+20}\right)\mathrm{mod}1$
   8:
$y_{0}=cos\left(\bar{m}\times 2^{y+20}\right)\mathrm{mod}1$
   9:
$\hat{h}=\left(cos\left(\bar{m}\times 2^{x_{0}\times y_{0}+20}\right)\mathrm{mod}1\right)\times 14+10$

### 3.2. S-Box Generation

#### 3.2.1. S-Box Generation Algorithm

In the construction method of a chaotic S-box, there is the problem of generating useless chaotic sequences that affect efficiency. Wang et al. proposed in [15] to use a three-dimensional chaotic map to iterate chaotic sequences and expand the modulo operation to generate values between 0 and 255 to generate S-boxes, and the algorithm only ends when all 256 values are generated, this method generates a large number of useless chaotic sequences, and if a value cannot be generated all the time, the efficiency of the algorithm is affected, and the real-time performance becomes poor. In some methods, there are immobile points in the generated S-boxes [29], which can become attackable points. Table 2 gives the generation times of existing S-box generation methods, the presence of fixed points and the number of fixed points before and after the Fisher–Yates shuffling algorithm is applied to the S-box.

The Fisher–Yates shuffling algorithm is an algorithm proposed by R. Fisher and F. Yates for generating random permutations of finite linear arrays [30]. The most important feature of this approach is that it generates an unbiased result so that the probability of each value being at any position is equally likely, essentially generating a finite set of random permutations. Thus, by using the Fisher–Yates random shuffle algorithm, we are able to make the encryption scheme more complex and secure by enabling us to quickly generate different S-boxes and reduce the presence of fixed points during each encryption. The steps are as follows:Step 1: Obtain the length m of the sequence P that needs to be shuffled;Step 2: Generate a random number n with a value between [0,m−1];Step 3: Shuffle the values of the two positions according to n and m, then exchange the values of P(n) and P(m);Step 4: Subtract 1 from m to obtain the new position;Step 5: Repeat step1~step4 until m=1.



Algorithm 2 describes the generation process of the S-box. $t_{0},\hat{\mu }$, generated by algorithm 1, is taken as the initial value and control parameter of Logistic, iteratively generating a chaotic sequence Q with a length of 512 bits, and then divided into two sequences Q1,Q2 with a length of 256 bits, respectively, calculating and obtaining the ascending index q1∈[0,255], q2∈[0,255] of Q1 and Q2. Then q2 is taken as the random number sequence of the shuffling algorithm, shuffling q1 and finally, q1 is converted into a 16 bits × 16 matrix to obtain the S-box s.
**Algorithm 2** Generation of S-box
Input: encryption keys
$t_{0},\hat{\mu }$
Output: S-box
s
   1: for
i from 1 to 1512:
       Substituting
$t_{0},\hat{\mu }$ into Equation (1)
       if
i >1000:
          obtain the 512-length chaotic sequences
Q
   2: Divide
Q into two subsequences of length 256 Q1, Q2; obtain the ascending 
    sort index of the Q1, Q2; and assign it to $q1=\arg sort\left(Q1\right)$ and $q2=\arg sort\left(Q2\right)$
   3: Read the size:
m=size(q1)
   4: while
m>1:
       Obtain a random number
$q2\left(m\right)$
       Swap the value of
$q1(q2\left(m\right)),q1(m)$
       Set
m=m−1
   5: If
m=1 transform q1 into 16×16 matrix and assign it to s
   6: Obtain S-box
s

#### 3.2.2. Performance Test of the Proposed S-Box

In order to verify the performance and strength of the generated S-boxes, we used the general criteria for S-box performance evaluation to be able to test them [31]. In this thesis, the nonlinearity of the S-box, the strict avalanche criterion SAC, the output bit independence criterion BIC and the differential approximate probability DP are verified, respectively. Table 3 displays the S-box matrix that we produced.

**(I) Nonlinearity** In the process of encryption, if the given S-box makes a linear mapping between the input (plaintext) and the output (ciphertext) [32], then a decipherer can easily deduce and break the ciphertext when the cryptographic strength of the S-box is very small, but if the S-box can map the input to the output in a nonlinear way, then it is considered a reliable S-box that can protect the plaintext data and can help us resist the attacks of linear cryptanalysis, and we can calculate the nonlinear value of the 8-bit Boolean function S by using Equation (13).
(13)$$NL_{f}=128-\frac{1}{2}\underset{\omega \in GF(2^{n})}{\max}\left|WH_{f}(\omega )\right|$$
where *NL**_f_* is the 8-bit Boolean function, $WH_{f}\left(\omega \right)$ is the Walsh–Hadamard transform of the eight-bit Boolean function S. The values of the S-box nonlinearity obtained according to the above are shown in Table 4, where the maximum value is 110, the minimum value is 106 and the average value is 107.5

**(II) Strict Avalanche Criterion (SAC)** Webster and Tavares used the strict avalanche criterion as an important feature of the performance of an S-box [33]. The strict avalanche criterion ensures that if one bit is changed in the input, it causes at least 50% of the output to change, and an S-box is considered to be a strong S-box if the value of SAC is approximately equal to 0.5. We propose that the S box has a SAC value that satisfies the strict avalanche criterion. The SAC dependency matrix of our generated S-boxes is shown in Table 5. The table shows the average SAC of the generated S-boxes is 0.4996, which is very close to the desired value of 0.5, and shows that the generated S-boxes satisfy the SAC criteria.

**(III) Output Bits Independence Criterion (BIC)** This criterion was introduced by Webster et al. as one of the important properties of S-box evaluation, a property that ensures that there is no dependence on the change of any two output bits when a single input bit is changed, a property that makes any Boolean function must be independent and highly nonlinear. For two output bit Boolean functions $f_{i}$ and $f_{j}$ in an S-box, if $f_{i}⊕f_{j}$ is highly nonlinear and satisfies the SAC as close as possible, then it is guaranteed that when one input bit is inverted, the correlation coefficient of each output bit is close to 0, i.e., the BIC is satisfied.

Through experimental tests, the average BIC nonlinearity value of the proposed S-box is 104.5, and the BIC-SAC test result is 0.5009, which satisfies the BIC criterion.

**(IV) Difference Approximation Probability (DP)** differential cryptanalysis, introduced by Biham and Shamir [34], is able to obtain the input differential from the output differential while being able to attempt to obtain from it modifications to the plaintext and changes to the ciphertext data, combined with the difference between the two changes an attacker is able to use the resulting small differences to identify complete or partial plaintexts and keys, and in the process of designing the S-box, there is a need to minimize both changes The difference between the two needs to be minimized in the design of the S-box. The designer calculates the difference by differential uniformity, which is checked right by the differential approximation probability, as shown in Equation (14):(14)$$DP(f)=\underset{\Delta x\neq 0,\Delta y}{\max}\left(\frac{\#\{x\in GF(2^{n})|f(x)⊕f(x+\Delta x)=\Delta y\}}{2^{n}}\right)$$
where Δx and Δy are the input difference and output difference, and DP denotes the maximum probability that the output of each given difference Δx is equal to Δy. The maximum DP value of our proposed generated S-box is 10, indicating that our S-box can resist differential.

In comparison with the other four methods, the results show in Table 6 that the S-box nonlinearity produced by the algorithm in this paper is higher than the other four solutions. The SAC value of S box 0.4996 is closest to the ideal value of 0.5 in five methods, and the BIC value also meets the test requirements. The BIC of the S-box also meets the test requirements based on the DP value of the test and has a good ability to resist the differential password attack based on the DP value of the test. In terms of S-box generation time, the algorithm in this paper has a good advantage. Although our nonlinear value is still some distance from the ideal value, we believe that the performance of the generated S-box needs to meet the performance requirements, and the efficiency and space cost of S-box generation also need to be considered.

### 3.3. Bit-Level Encryption

Traditional image encryption generally falls into one of three categories: permutation-only, diffusion-only, or combined forms. Due to its simpler computational complexity, the permutation-only type of these is more efficient, but the security is not very strong. Due to the substantial computational load required for real arithmetic operations, the diffusion type is a time-consuming process. The pixel values and random numbers are used to perform an XOR operation, and the calculated numbers are used to determine the ranks of the S-boxes and substitute the original pixels with the values in the S-boxes, but the overall encryption efficiency is low due to the lack of permutation and diffusion operations on the plaintext image and the low efficiency of S-box generation [15].

In this stage, by separating the eight-bit pixel values into upper four bits and lower four bits, different operations are performed on the two parts, respectively, and a new diagonal diffusion method for irregular matrices is proposed to be applied to the high four-bit matrix. The bit-level encryption process is shown in Figure 8.

#### 3.3.1. Pixel Value Split

We converted the original pixel value of the plaintext image into a binary representation and extracted it into two parts for different operations. The upper four bits P1 of the pixel value and the lower four bits P2 of the pixel value are shown in Figure 9.

#### 3.3.2. Improved Diagonal Diffusion

In order to make the information of pixel points have a good diffusion effect, many diffusion methods, such as V-diffusion, zigzag diffusion and chunk diffusion, were proposed. A zigzag algorithm was used in [35] and improved to rearrange the pixel values, but the transformation law of zigzag is relatively single, traversing from the upper left foot to the lower right corner in a Z-shaped order, as shown in Figure 10.

However, the security of the zigzag is not guaranteed due to the fact that it is extremely easy to be broken by a single variation.

In order to obtain faster diffusion speed and diffusion effect, this paper proposes a new diagonal diffusion method by randomly and irregularly rearranging the image matrix through a chaotic sequence. The number of pixel values in each row is controlled by the random sequence, there may be no pixel values at some positions of the diagonal, and the matrix presents an irregular arrangement. Through the proposed new diagonal diffusion, it can complete the diagonal of diffusion. Through the comparative analysis of the relevant experiments in Section 4, this scheme has good performance.

In this paper, after obtaining the upper four-bit matrix, we changed its alignment through a random sequence to generate different irregular matrices, performed diagonal diffusion on the matrix according to the changed alignment and changed its alignment, as shown in Figure 11. Algorithm 3 describes the transformation process.
**Algorithm 3** Matrix rearrange
Input: upper 4-bits sequence
P1, random sequence $X^{\prime }(i)$
Output: irregular matrices
$P1^{\prime }$
   1: Obtain the size of
P1, m = size (P1)
   2: Set an empty list
$P1^{\prime }$ and i=0,j=0
   3: While
$\left(m-X^{\prime }(i)>0\right)$:
          $P1^{\prime }(i)=P1(j:j+X^{\prime }(i))$
          i=i+1
      $j=j+X^{\prime }\left(i\right)$
      $m=m-X^{\prime }\left(i\right)$
   4:
$P1^{\prime }\left(j\right)=P1\left(j:\right)$

Where $P1\left(i:j\right)$ denotes from the ith to the jth digit of P1, and $P1\left(i:\right)$ denotes from the ith to the last digit of P1.

Based on the rearranged matrices, we proposed a diagonal diffusion to perform diffusion operations on upper 4-bit matrices capable of new diagonal diffusion operations based on different irregular matrices, as shown in Figure 12, where the arrows represent the order and direction.

According to the irregular matrix obtained by Algorithm 3, we need to diffuse according to the diagonal, but the shape of the matrix is irregular, so given Algorithm 4 used to obtain the order of the pixel values of the diagonal of the irregular matrix, sequentially traversing the diagonal to obtain its diffusion order matrix for diffusion, according to different irregular matrices for the order of diagonal diffusions, such as the first: (1- > 3- > 2- > 7- > 4- > 8- > 5- > 9- > 6), the second: (1- > 2- > 4- > 3- > 5- > 6- > 7- > 8- > 9) and the third: (1- > 5- > 2- > 6- > 3- > 9- > 7- > 4- > 8), if we use the normal zigzag, then the order of each diffusion is fixed in a 3 × 3 matrix, and the result is (1- > 2- > 4- > 7- > 5- > 3- > 6- > 8- > 9) each time, which is not guaranteed in terms of security.
**Algorithm 4** Diagonal diffusion order
Input: irregular matrices
$P1^{\prime }$
Output: order sequence
D
   1: Obtain the size of
$P1^{\prime }\;m$ = size $\left(P1^{\prime }\right)$
   2: Set tow empty list
D, sub
   3: for
i from 1 to m:
       Obtain the size of
$P1^{\prime }\;n=size(P1^{\prime }(i))$
       for
i from 1 to n:
          $sub(i+j).insert\left(0,P1^{\prime }\left(i,j\right)\right)$
   4: Obtain the size of
sub m=size(sub)
   5: for
i from 1 to m:
       Obtain the size of
$sub\left(i\right)\;n=\mathrm{size}\;(sub\left(i\right))$)
       for i from 1 to n:
           $D.\;insert\left(sub\left(i,j\right)\right)$

Where $sub(i+j).insert\left(0,P1^{\prime }\left(i,j\right)\right)$ means adding $P1^{\prime }\left(i,j\right)$ to the head of the array $sub\left(i+j\right)$ and $D.\;insert\left(sub\left(i,j\right)\right)$ means adding sub(i,j) to the end of the array D.

#### 3.3.3. Permutation and Diffusion Process


Step 1: Generate S-boxes s according to the S-box generation method proposed in Section 3.2.;Step 2: Generate the encryption key $\hat{\theta },x_{0},y_{0},\hat{h}$ and process the image to obtain P1,P2 according to Algorithm 1 and θ,x,y;Step 3: Iterate over $\hat{\theta },x_{0},y_{0},\hat{h}$ as the control parameter and initial value of the 2D-CLSM chaos map to generate two chaotic sequences $X=\{x_{1},x_{2},\ldots ,x_{M\times N}\},Y=\{y_{1},y_{2},\ldots ,y_{M\times N}\}$ of length M×N;Step 4: Process the chaotic sequence to obtain the random numbers for array rearrangement;
(15)$$X^{\prime }\left(i\right)=\left(floor\left(X\left(i\right)\times 10^{13}\right)\mathrm{mod}N\right)+1,\;i=1,2,\ldots ,M\times N$$Step 5: Based on the resulting upper 4-bit matrix P1 and the random sequence $X^{\prime }\left(i\right)$, the upper 4-bit matrix is rearranged by Algorithm 3 to obtain a new irregular matrix $P1^{\prime }$;Step 6: Perform a diffusion operation on the irregular matrix $P1^{\prime }$ obtained in step 4. For computational convenience, we first transform the irregular matrix into a one-dimensional matrix D by means of Algorithm 4;Step 7: Diffusion operation based on the 1D matrix D obtained in step 6 to obtain $D^{\prime }$;
(16)$$\left\{\begin{array}{l} D^{\prime }\left(i\right)=\left(X^{\prime }\left(1\right)\mathrm{mod}16\right)⊕D\left(i\right),\;i=1\\ D^{\prime }\left(i\right)=D^{\prime }\left(i-1\right)⊕D\left(i\right),\;i=2,3,\ldots ,M\times N \end{array}\right.$$Step 8: Process the chaotic sequence to obtain chaotic values for the lower four positions;
(17)$$Y^{\prime }\left(i\right)=floor\left(Y\left(i\right)\times 10^{13}\right)\mathrm{mod}65536\;i=1,2,\ldots ,M\times N$$Step 9: Based on the resulting $Y^{\prime }\left(i\right)$ the lower four bits of the matrix are permutated.
(18)$$\left\{\begin{array}{l} temp=P2\left(Y^{\prime }\left(i\right)\right)\\ P2\left(Y^{\prime }\left(i\right)\right)=P2\left(i\right)\\ P2\left(i\right)=temp \end{array}\right.,\;i=1,2,\ldots ,M\times N$$



#### 3.3.4. Pixel-Level Encryption

At the end of the bit-level encryption, pixel-level encryption is performed by the resulting pixel values using the S-box. It mainly includes pixel value diffusion and substitution. The main process is shown in Figure 13.

Generate an empty matrix C of size M×N. Based on the matrix $D^{\prime },P2$ after completing the diffusion and permutation, reorganize $D^{\prime },P2$ by replacing the original upper four bits as lower four bits and the original lower four bits as upper four bits to obtain 8-bit values between 0 and 255, calculate the row index r and column index c of the S-box and complete the pixel value substitution using the S-box s generated in the second stage.
(19)$$\left\{\begin{array}{l} temp=\mathrm{dec}\left(\left\lceil P2\left(i\right)+D^{\prime }\left(i\right)\right\rceil \right)⊕X^{\prime }\left(i\right)⊕C(i-1)\\ c=temp\%16\\ r=\left(temp-c\right)/16\\ C(i)=s(r,c) \end{array}\right.,\;i=1,2,\ldots ,M\times N$$
where i=1, C(i−1)=0; $i=M\times N,\;C(i-1)=C\left(1\right)$. $\mathrm{dec}\left(\left\lceil a+b\right\rceil \right)$ is the reconfiguration of two four-bit binary numbers a,b into an eight-bit binary number, and s(r,c) represent obtaining the value at row r and column c from the S-box s.

Finally, C is converted to an M×N ciphertext image.

## 4. Simulation Experiments

The simulation results of the proposed method are presented in this section by running PyCharm software under Windows 10 64-bit system. By using the method in this paper, 10 images were both encrypted and decrypted. The results of encrypting and decrypting several grey-scale images such as Baboon.png, House.png, Cameraman.png and Peppers.png pixels are shown here, respectively. The initial keys used for encryption/decryption are shown in Table 7, and generated keys are shown in Table 8, where $t_{0},\hat{\mu }$ are used as initial values and control parameters for the Logistic. $x_{0},y_{0},\hat{h},\hat{\theta }$ are used as control parameters, initial values and variables h for the 2D-CLSM.

From Table 8, we can clearly see that the encryption keys generated by the algorithm in this paper have good plaintext correlation, can generate completely different encryption keys depending on the plaintext, have good plaintext sensitivity and can effectively resist selective plaintext/known plaintext attacks.

Figure 14a–c shows the experimental results of Baboon, Figure 14d–f shows the experimental results of House, Figure 14g–i shows the experimental results of Cameraman and Figure 14j–l shows the experimental results of Peppers. It is clear from the Figure 14 that the encrypted image can still be fully restored, which verifies the effectiveness of the algorithm in this paper, which can obtain good encryption and decryption results.

### 4.1. Security Analysis

#### 4.1.1. Key Space Analysis

Assuming that the accuracy of the computer is $10^{-16}$, in order to make the key sufficiently resistant to brute-force attacks, the key space of the encryption system must be more than $2^{100}$, and the parameters of the encryption system are $t_{0},\hat{\mu },\hat{\theta },x_{0},y_{0},\hat{h}$, $t_{0}\in [0,1],\hat{\mu }\in [0.89,1],\hat{\theta }\in [0,1],x_{0}\in [0,1],y_{0}\in [0,1],\hat{h}\in [10.24]$, so the key space is calculated as in Equation (20). The key space is larger than $2^{100}$, so the algorithm in this paper has sufficient key space to resist brute-force cracking attacks.
(20)$$0.11\times 10^{16}\times 10^{16}\times 10^{16}\times 10^{16}\times 10^{16}\times 14\times 10^{16}>2^{100}$$

#### 4.1.2. Key Sensitivity Analysis

In a secure encryption system, a mirror change in the key can cause a complete change in the ciphertext obtained from the encryption. Through testing, the encryption key t,μ of the Peppers graph was increased by $10^{-16}$ to the original one, respectively, and the rest of the keys were left unchanged, and the Peppers were encrypted with the modified key and the original key, respectively, and the results obtained are shown in Figure 15a–f.

As shown in Figure 15, when the initial key is slightly changed, the resulting encrypted image is completely changed, which shows that the algorithm has good key sensitivity and only the correct key can decrypt the ciphertext.

#### 4.1.3. Histogram Analysis

The distribution of the image’s pixel values can be directly described by the histogram. A secure encryption system should ensure that pixel values of the obtained ciphertext image are uniformly distributed in order to reduce readability, improve security and provide effective protection against statistical attacks.

Figure 16 shows the histogram of plaintext and ciphertext pixel frequencies for Baboon, House, Cameraman and Peppers, from which we can see that the pixel of ciphertext images is uniformly distributed. The statistical properties of the images were altered so that they are well resistant to statistical analysis attacks.

The variance of the histogram can be used to specifically describe the distribution of pixel values. The calculation equation is shown in (21), and the result shows in Table 9.
(21)$$var(z)=\frac{1}{N^{2}}\sum_{i=0}^{N-1}\sum_{j=0}^{N-1}\frac{1}{z}{\left(z_{i}-z_{j}\right)}^{2}$$

Where z represents histogram values; N represents the total number of samples (N for an image with a grey level of 8 is 256); $z_{i}$, $z_{j}$ are number of pixels with grey values at i, j. The smaller the histogram’s variance, the more evenly distributed the histogram.

Table 9 compares the variance of histograms with methods in Refs. [18,19,20,21,22], from which we can see that our encrypted Baboon, House, and Cameraman have the lowest variance of all the schemes, and Peppers is slightly higher than the others. The average variance of the four images is 236.26, which is the lowest of the four methods, so it performs better in resisting statistical attacks.

#### 4.1.4. Plaintext Sensitivity Analysis

A differential attack is a common form of attack in cryptography, in which an attacker usually selects to encrypt the plaintext and a slightly altered plaintext and analyses them to find a specific relationship. In order to be effective against differential attacks, the encryption algorithm should be sensitive enough to the plaintext that a diffusion operation during encryption can cause small changes in the plaintext to affect all pixels, and we usually evaluate the sensitivity of the algorithm using the pixel change rate NPCR and the normalized pixel average change intensity UACI, their ideal expectations are 99.6094% and 33.4635%, respectively. We obtained NPCR and UACI values by randomly modifying two pixel points of the plaintext image. In this paper, we choose to modify pixel points (2, 2) and (236, 207). In Table 10, we give the corresponding experimental results as well as the corresponding results for the other four scenarios. The calculation expressions are shown in Equations (22)–(24).
(22)$$NPCR=\frac{1}{M\times N}\sum_{i=0}^{M-1}\sum_{j=0}^{N-1}D(i,j)\times 100\%$$
(23)$$D(i,j)=\left\{\begin{array}{l} 0,C(i,j)=C^{\prime }(i,j)\\ 1,C(i,j)\neq C^{\prime }(i,j) \end{array}\right.$$
(24)$$UACI=\frac{1}{M\times N}\sum_{i=0}^{M-1}\sum_{j=0}^{N-1}\frac{\left|C_{1}(i,j)-C_{2}(i,j)\right|}{255}\times 100\%$$
where i=0,1,…,M−1,j=0,1,…,N−1, C, $C^{\prime }$ are original encrypted image and the encrypted image with the plaintext modified, respectively; M and N make the height and length of the image, respectively; and $C\left(i,j\right)$ denotes the pixel values at coordinates $\left(i,j\right)$ of the encrypted image C.

Table 10 compares the NPCR and UACI values of our scheme with the methods in Refs. [18,19,20,21,22]. In our method, the NPCR and UACI values of Baboon, House and Cameraman images exceed the expected values, and the NPCR values of Peppers are close to the expected values. Of the other methods, only the Cameraman plot of Ref. [20] achieved the desired values for both NPCR and UACI values. In contrast, our encryption algorithm has better sensitivity to plaintext.

#### 4.1.5. Correlation Analysis

When the image is not encrypted, there is a strong correlation between the pixel values of the images; the encryption operation on the plaintext allows us to break this correlation. We calculated the correlation coefficient by randomly selecting n pair of adjacent pixels $\left(a_{i},b_{i}\right)$ from the image for which the correlation is to be calculated using the following Equation (25). Table 11 shows the experimental results for a random selection of 3000 pairs of pixel values.
(25)$$r_{ab}=\frac{\sum_{i=1}^{n}\left(a_{i}-\bar{a})(b_{i}-\bar{b}\right)}{\sqrt{\sum_{i=1}^{n}{\left(a_{i}-\bar{a}\right)}^{2}}\sqrt{\sum_{i=1}^{n}{\left(b_{i}-\bar{b}\right)}^{2}}}$$
where $r_{ab}$ represents the correlation coefficients, $\bar{a}$, $\bar{b}$ is the mean value.

The closer the calculated correlation coefficient is to 1, the stronger the correlation is, and the closer it is to 0, the weaker the correlation is. As shown in Table 11, take baboon as an example; we are able to see in Table 11 that it has a horizontal correlation of 0.0007, a vertical correlation of −0.003 and a diagonal correlation of 0.008, with a correlation index very close to 0 in three directions, indicating that the high correlation between pixels is broken.

In addition, from Figure 17, we can clearly see that for the plaintext image, most of the points are close to the diagonal of the axes, whereas, for the encrypted image, these points are randomly distributed throughout the space, with significantly lower inter-pixel correlation. Both illustrate the effectiveness of our algorithm in removing intra-pixel correlations.

#### 4.1.6. Information Entropy Analysis

Information entropy is another metric for evaluating the security of a ciphertext and represents the strength of the uncertainty of the image information, which is expressed in Equation (12).

For a grey-scale image with 256 states, it has a maximum expectation value of 8, Table 12 shows the test results we obtained and the comparison of the other four schemes, where the result of the three encrypted images of House, Baboon and Peppers are higher than the other schemes, and the results of all four images are close to the ideal value of 8. It can be said that the image results processed by our encryption scheme are close to the random images, with good uncertainty, and can effectively resist statistical analysis attacks.

#### 4.1.7. Anti-Cropping Attack Analysis

During the transmission of information, information may be lost due to humans or some uncontrollable factors. If the encryption algorithm is able to restore the plaintext image in this case, then the encryption algorithm is valid. In Figure 18, we add a 5% loss of pixel values to the Peppers. From the perspective of the decrypted image, even though the image has lost some data, it is still possible to decrypt the basic information and be able to recover it basically, so the algorithm is resistant to basic cropping attacks.

#### 4.1.8. Speed Analysis

Running speed is an important index to evaluate the encryption algorithm; we simulated it in Python 3.7 environment on a PC with a 2.2 GHz CPU and 8 G RAM. As can be seen from Table 13, this paper’s encryption efficiency is superior to that of Refs. [11,12,13] while being slightly slower than that of Ref. [15]. This shows that our proposed encryption algorithm performs well in terms of efficiency and is capable of being applied in practical applications. Because we used a round of bitrate encryption and a round of pixel-level encryption, the efficiency is reduced, but the security is improved.

## 5. Conclusions

In this paper, we propose an improved cascaded chaotic map 2D-CLSM and design a novel encryption scheme based on it and S-box. By generating a key associated with the plaintext, the whole encryption process is associated with the plaintext, effectively resisting the chosen/known plaintext attack. An S-box encryption scheme is designed and applied to our encryption, increasing the overall security of the algorithm. Dividing the encryption into bit-level encryption and pixel-level encryption improves the complexity and security of the encryption to a certain extent but reduces its efficiency. The encryption method in this paper can be applied to different image types, such as grey-scale images, grey-scale medical images, etc., while the algorithm is able to meet everyday encryption requirements, both in terms of efficiency and security. The experimental results show that the algorithm has sufficient key space, is resistant to brute force attacks and performs well against statistical analysis attacks, clipping attacks and differential attacks. However, although we improved the efficiency of S-box generation, we used two rounds of encryption, one at the bit level and one at the pixel level, which makes us less efficient overall, so the next step is to improve the efficiency of encryption in both rounds.

## Figures and Tables

**Figure 1 entropy-24-01827-f001:**
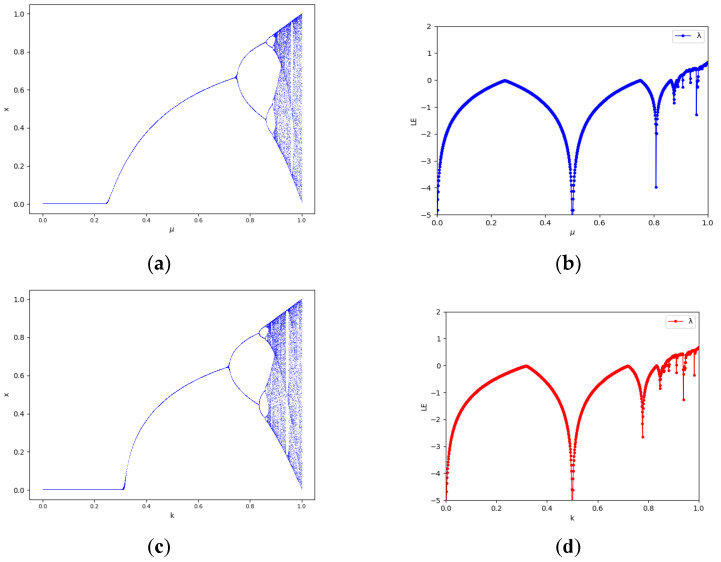
Chaotic bifurcation diagrams and Lyapunov exponents: (**a**) bifurcation diagram of Logistic map; (**b**) Lyapunov exponent of Logistic map; (**c**) bifurcation diagram of Sine map; (**d**) Lyapunov exponent of Sine map.

**Figure 2 entropy-24-01827-f002:**
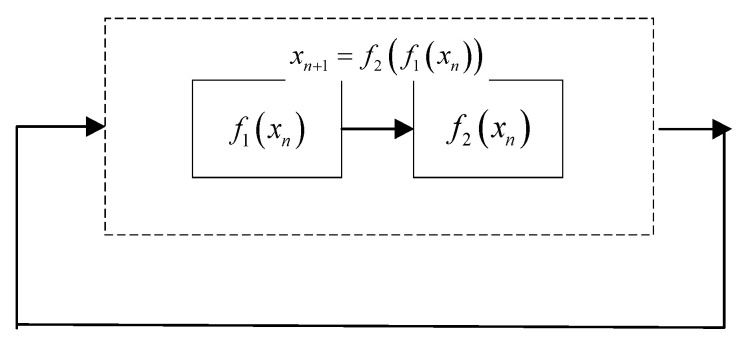
Cascade system.

**Figure 3 entropy-24-01827-f003:**
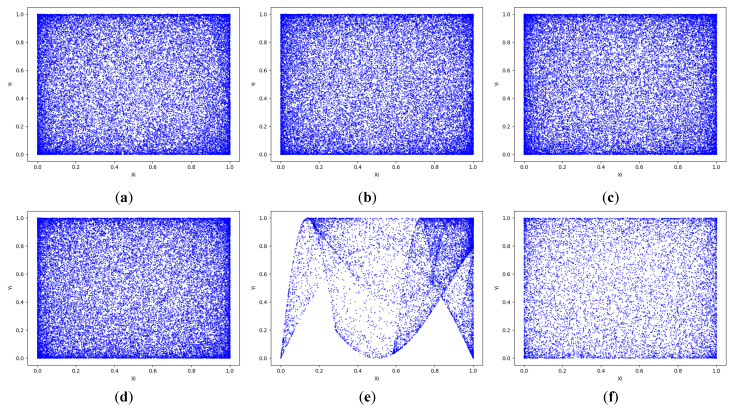
The trajectory of chaotic systems: (**a**) the trajectory of 2D-CLSM with θ=0.71,h=15; (**b**) the trajectory of 2D-CLSM with θ=0.3,h=15; (**c**) the trajectory of 2D-CLSM with θ=0.71,h=20; (**d**) the trajectory of 2D-CLSM with θ=0.3,h=20; (**e**) the trajectory of 2D-LSCM with θ=0.3; (**f**) 2D-LSCM with θ=0.71.

**Figure 4 entropy-24-01827-f004:**
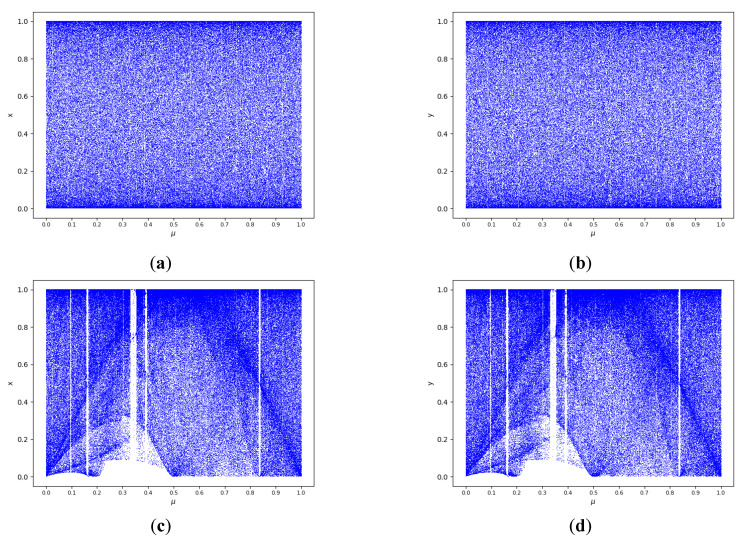
Bifurcation diagram of chaotic system: (**a**) bifurcation diagram of 2D-CLSM θ−x; (**b**) bifurcation diagram of 2D-CLSM θ−y; (**c**) bifurcation diagram of 2D-LSCM θ−x; (**d**) bifurcation diagram of 2D-LSCM θ−y.

**Figure 5 entropy-24-01827-f005:**
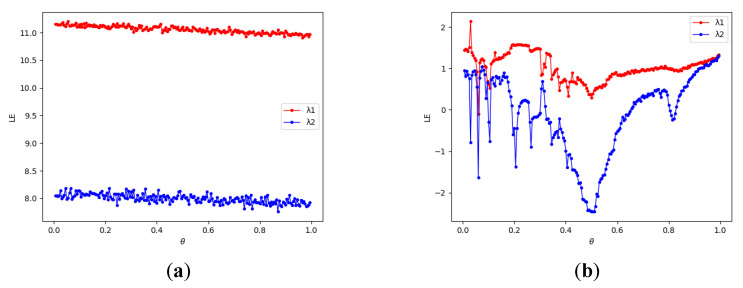
Lyapunov exponents for chaotic systems: (**a**) Lyapunov exponent of 2D-CLSM; (**b**) Lyapunov exponent of 2D-LSCM.

**Figure 6 entropy-24-01827-f006:**
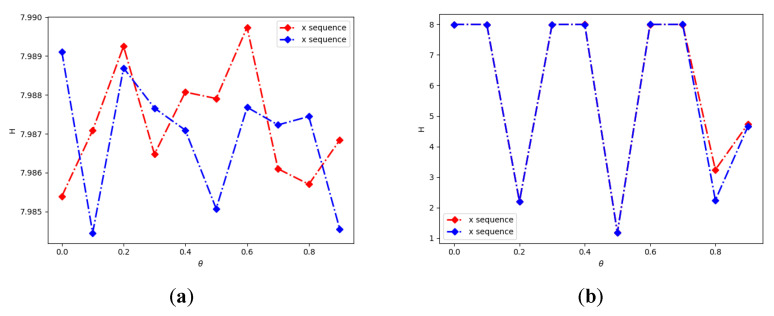
The information entropy of the sequences: (**a**) information entropy of 2D-CLSM; (**b**) information entropy of 2D-LSCM.

**Figure 7 entropy-24-01827-f007:**
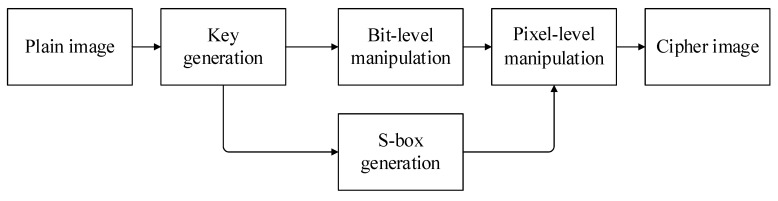
Overall process.

**Figure 8 entropy-24-01827-f008:**
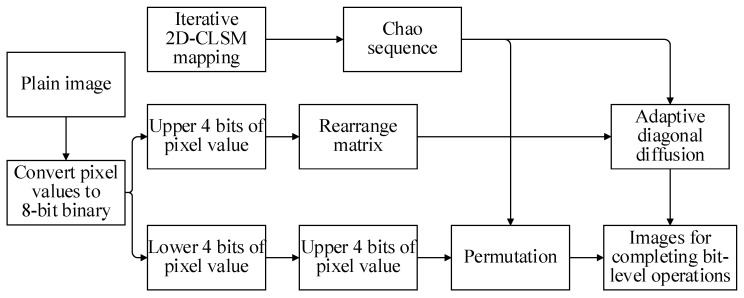
Bit-level encryption process.

**Figure 9 entropy-24-01827-f009:**
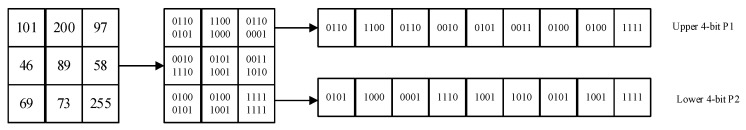
Plaintext image processing.

**Figure 10 entropy-24-01827-f010:**
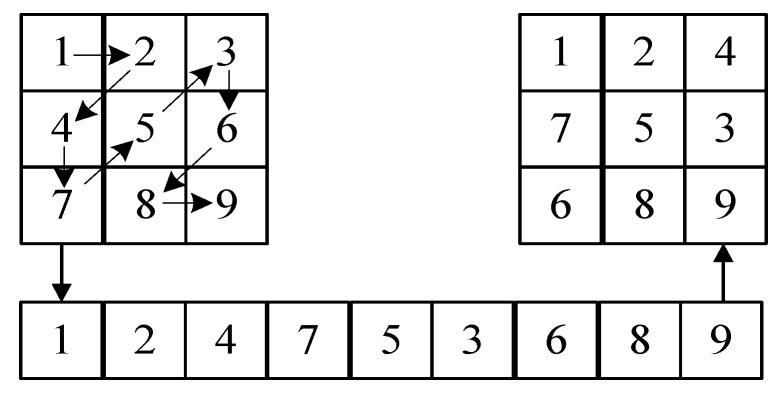
Zigzag transform.

**Figure 11 entropy-24-01827-f011:**
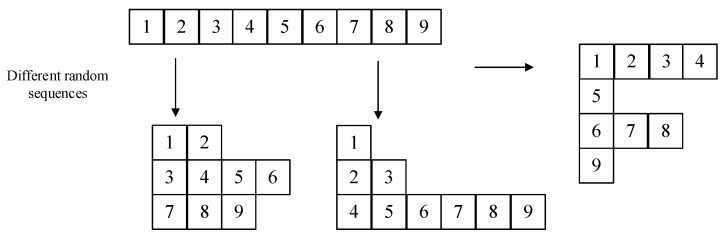
Matrix rearrangement.

**Figure 12 entropy-24-01827-f012:**
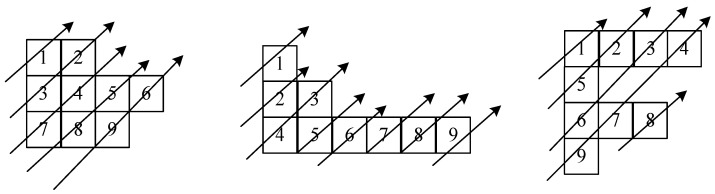
Diagonal diffusion of different matrices.

**Figure 13 entropy-24-01827-f013:**
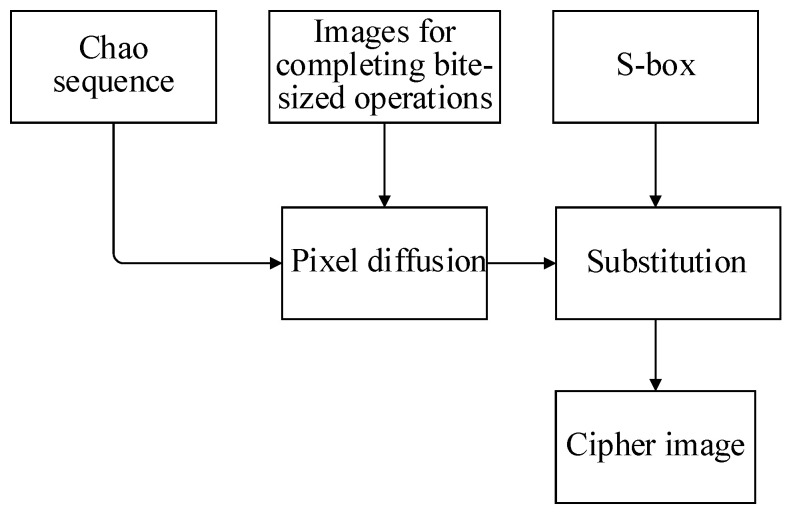
Pixel-level encryption process.

**Figure 14 entropy-24-01827-f014:**
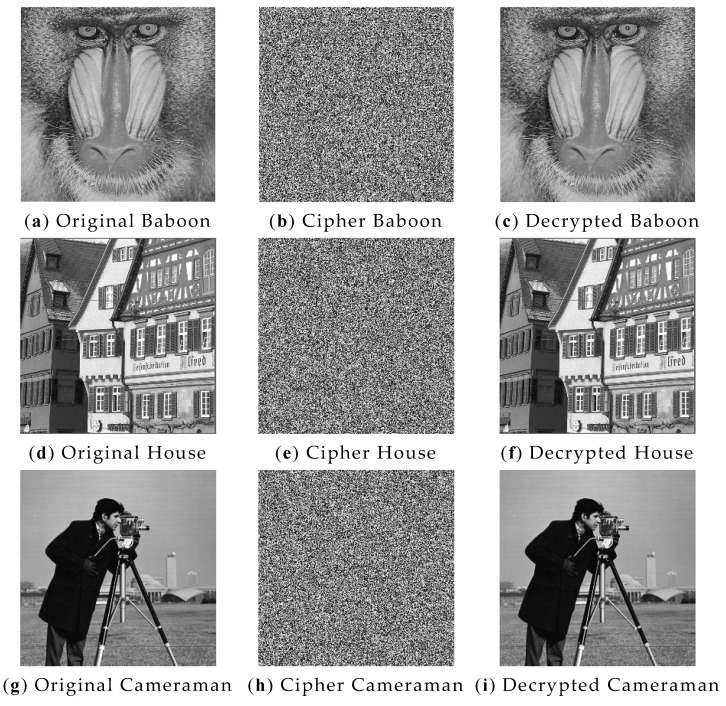
Encryption and decryption results: (**a**) plaintext image Baboon; (**b**) encrypted image Baboon; (**c**) decrypted image Baboon; (**d**) plaintext image House; (**e**) encrypted image House; (**f**) decrypted image House; (**g**) plaintext image Cameraman; (**h**) encrypted image Cameraman; (**i**) decrypted image Cameraman; (**j**) plaintext image Peppers; (**k**) encrypted image Peppers; (**l**) decrypted image Peppers.

**Figure 15 entropy-24-01827-f015:**
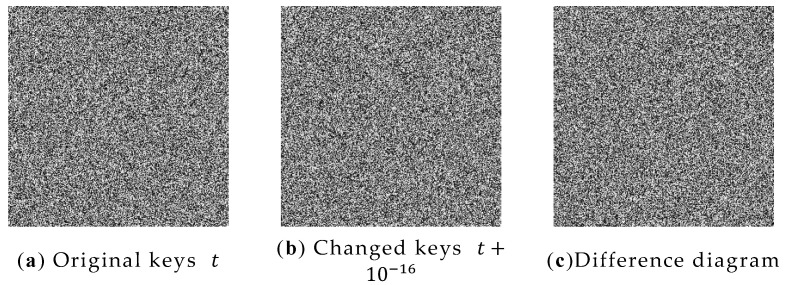
Key sensitivity analysis: (**a**) image encrypted with original key t; (**b**) image encrypted with original key $t+10^{-16}$; (**c**) the difference between (**a**,**b**); (**d**) image encrypted with original key μ; (**e**) image encrypted with original key $\mu +10^{-16}$; (**f**) the difference between (**d**,**e**).

**Figure 16 entropy-24-01827-f016:**
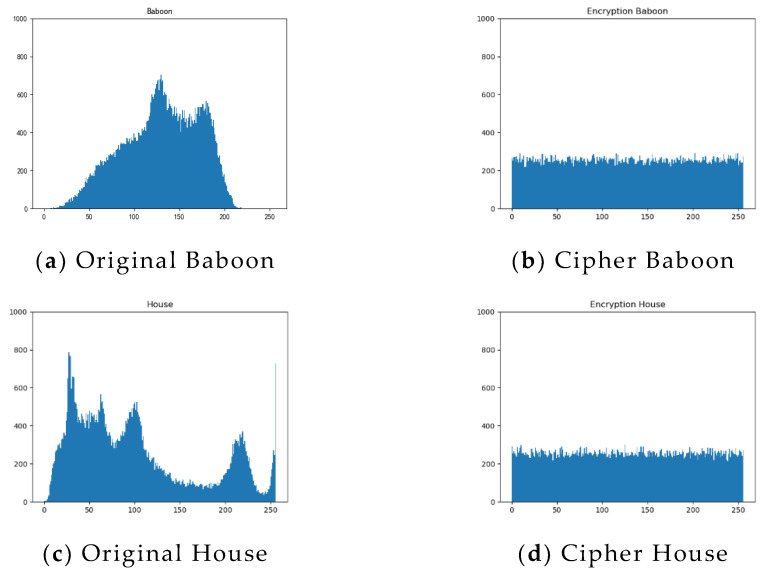
Plain text image histogram and Ciphertext image histogram analysis: (**a**) plaintext Baboon histogram; (**b**) Ciphertext Baboon histogram; (**c**) plaintext House histogram; (**d**) Ciphertext House histogram; (**e**) plaintext Cameraman histogram; (**f**) Ciphertext Cameraman histogram; (**g**) plaintext Peppers histogram; (**h**) Ciphertext Peppers histogram.

**Figure 17 entropy-24-01827-f017:**
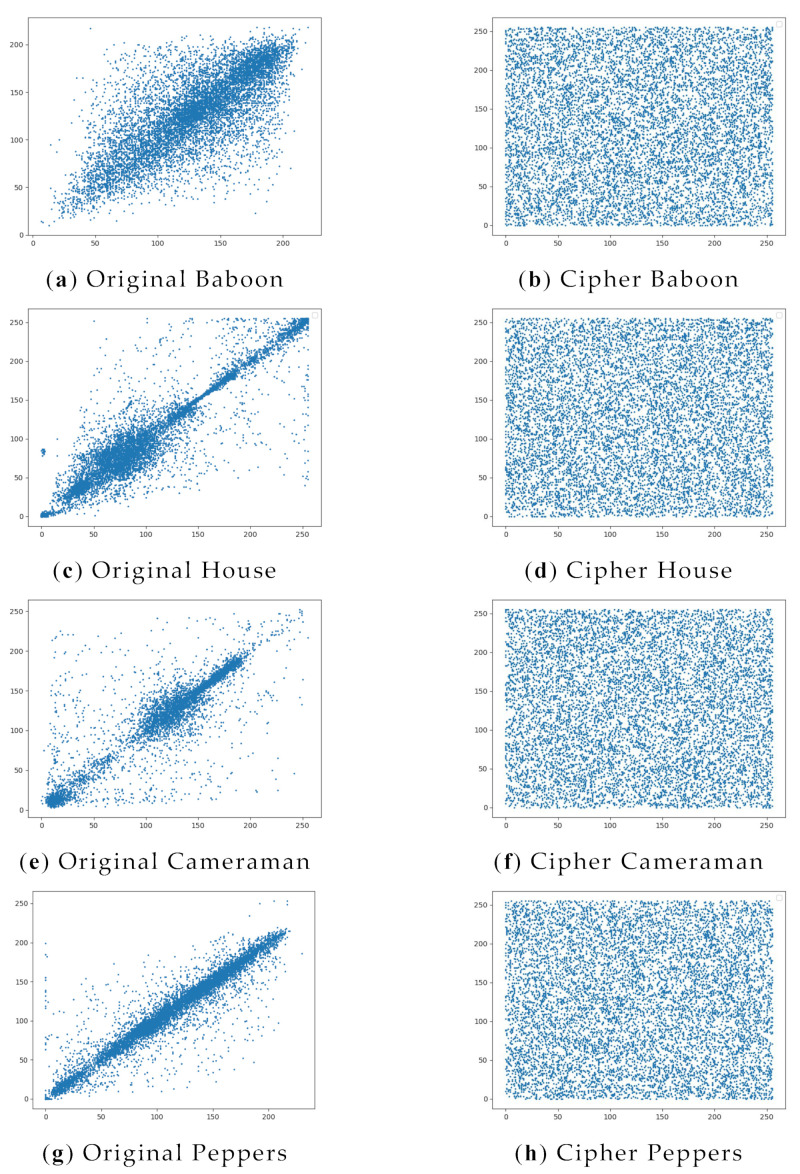
Plain text image correlation and Ciphertext image correlation analysis: (**a**) plaintext Baboon diagonal direction; (**b**) Ciphertext Baboon diagonal direction; (**c**) plaintext House diagonal direction; (**d**) Ciphertext House diagonal direction; (**e**) plaintext Cameraman diagonal direction; (**f**) Ciphertext Cameraman diagonal direction; (**g**) plaintext Peppers diagonal direction; (**h**) Ciphertext Peppers diagonal direction.

**Figure 18 entropy-24-01827-f018:**
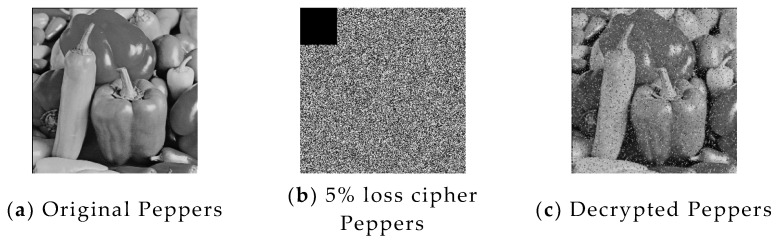
The cropped encrypted image and the corresponding decrypted image. (**a**) The plain image of Peppers. (**b**) The cipher image of Peppers with 5% loss. (**c**) The decrypted image of (**b**).

**Table 1 entropy-24-01827-t001:** NIST test results.

Serial Number	Test Items	2D-CLSM		2D-LSCM	
		*p* Value	Test Results	*p* Value	Test Results
1	Frequency	0.1422	Success	0.0767	Success
2	Block Frequency	0.7165	Success	0.9936	Success
3	Cumulative Sums	0.2472	Success	0.0692	Success
4	Runs	0.8561	Success	0.7405	Success
5	Longest Run of Ones	0.7310	Success	0.4477	Success
6	Rank	0.1691	Success	0.1514	Success
7	Discrete Fourier Transform	0.8330	Success	0.1766	Success
8	Nonperiodic Template Matchings	0.6254	Success	0.4721	Success
9	Overlapping Template Matchings	0.6886	Success	0.9365	Success
10	Universal	0.9992	Success	0.9583	Success
11	Approximate Entropy	0.9309	Success	0.7040	Success
12	Random Excursions	0.1319	Success	0.2175	Success
13	Random Excursion Variant	0.1025	Success	0.4166	Success
14	Serial	0.9068	Success	0.1638	Success
15	Linear Complexity	0.9250	Success	0.5041	Success

**Table 2 entropy-24-01827-t002:** Generation time and fix point.

S-Box	Generate Time	Fixed Point	After Fisher–Yates
Ref. [17]	0.9415	2, 17, C7, CB	None
Ref. [21]	0.0487	0D, 33, 77, 95	None
Ref. [22]	0.7593	None	None

**Table 3 entropy-24-01827-t003:** The generated S-box by proposed algorithm.

i\j.	0	1	2	3	4	5	6	7	8	9	A	B	C	D	E	F
0	100	62	60	203	217	27	159	103	77	112	134	236	2	167	219	96
1	228	29	18	170	113	39	64	127	87	90	1	160	94	183	7	125
2	199	54	55	193	104	246	146	129	79	14	162	137	237	63	191	174
3	148	115	109	99	225	13	202	187	17	250	185	41	110	25	139	177
4	50	15	76	238	114	34	12	107	207	222	45	102	249	75	220	200
5	98	195	47	0	151	51	67	3	82	230	184	204	241	117	35	130
6	36	254	156	196	227	178	248	68	145	31	126	149	153	5	43	181
7	19	157	30	154	121	231	86	201	239	101	189	72	69	71	119	37
8	131	11	118	10	83	24	215	140	247	74	38	152	46	206	8	106
9	59	208	164	150	136	192	255	9	84	235	229	88	213	171	147	92
A	166	48	97	28	224	180	23	144	85	4	190	122	111	173	108	243
B	52	188	210	209	197	58	182	233	143	40	26	163	33	244	218	211
C	120	89	20	16	124	57	53	232	142	179	73	172	22	44	175	70
D	176	212	32	216	91	194	49	245	155	80	161	234	141	42	226	198
E	186	135	205	61	240	123	223	251	105	21	95	133	253	221	252	242
F	66	93	169	116	81	165	78	128	138	132	214	56	65	6	168	158

**Table 4 entropy-24-01827-t004:** S-box nonlinear values.

	S1	S2	S3	S4	S5	S6	S7	S8
NL(s)	108	106	108	106	108	108	110	106

**Table 5 entropy-24-01827-t005:** Generating S-box SAC values.

0.4196	0.4235	0.4196	0.5647	0.4823	0.5137	0.5450	0.5294
0.4549	0.5019	0.4509	0.5019	0.5490	0.4705	0.4352	0.5176
0.5960	0.4862	0.4980	0.5137	0.5607	0.5333	0.5607	0.5176
0.4980	0.5137	0.5490	0.4980	0.4392	0.4862	0.5450	0.5490
0.5019	0.5294	0.5294	0.4980	0.4392	0.4862	0.5450	0.5490
0.5019	0.4509	0.5647	0.5176	0.5137	0.4980	0.5137	0.4392
0.4666	0.5333	0.4823	0.4235	0.4549	0.5450	0.5294	0.4235
0.4705	0.5333	0.4823	0.4549	0.5490	0.5294	0.5019	0.4980

**Table 6 entropy-24-01827-t006:** S-box performance test results.

S-Box	Nonlinearity			SAC	BIC-SAC	BIC-Nonlinearity	DP	Generate Time
	Min	Max	Avg					
ours	106	110	107.5	0.4996	0.5009	104	10	0.0066
Ref. [18]	100	106	104	0.4988	0.5006	104	10	0.3071
Ref. [19]	102	108	104	0.4988	0.5052	104	10	0.0091
Ref. [20]	106	108	106	0.4916	0.5058	104.14	10	0.0160
Ref. [22]	99	106	103.5	0.5065	0.5013	103.357	12	0.7593

**Table 7 entropy-24-01827-t007:** Initial key.

t	μ	θ	x	y	Δm
0.9	0.5	0.2	0.5	0.2	5001

**Table 8 entropy-24-01827-t008:** Generated encryption keys.

Image	Baboon	House	Cameraman	Peppers
$t_{0}$	0.1202650498467574	0.9477909907361988	0.2160499006239973	0.0961937648047185
$\hat{\mu }$	0.9349678430636638	0.9505574749633823	0.9210123850695779	0.9032945073229159
$\hat{\theta }$	0.0664003944646814	0.6920508714375233	0.2309975836114158	0.977565451998436
$x_{0}$	0.0264462005880243	0.4490902428109265	0.1811279735456482	0.1440190683132808
$y_{0}$	0.0664003944646814	0.6920508714375233	0.2309975836114158	0.977565451998436
$\hat{h}$	20.512754134641057	16.49739814853646	22.592144200059774	11.400485742889497

**Table 9 entropy-24-01827-t009:** Variance of histograms of encrypted images.

Image	Plain Image	Ours	Ref. [18]	Ref. [19]	Ref. [20]	Ref. [22]
Baboon	47,065.25	233.35	270.33	270.68	271.31	237.31
House	28,706.41	217.75	255.35	263.38	246.16	246.76
Cameraman	10,5149.27	247.92	269.91	249.93	265.32	259.79
Peppers	35,550.14	246.04	234.49	242.46	240.74	242.06
Average	54,513.19	236.26	257.52	256.61	255.88	246.48

**Table 10 entropy-24-01827-t010:** NPCR and UACI values.

Image		NPCR	UACI
Baboon	Ours	99.6246%	33.4949%
	Ref. [18]	99.4863%	32.1574%
	Ref. [19]	99.5616%	33.0145%
	Ref. [20]	99.6551%	33.4146%
	Ref. [22]	99.5256%	33.3324%
House	Ours	99.6292%	33.5274%
	Ref. [18]	99.4515%	33.1501%
	Ref. [19]	99.5849%	33.3829%
	Ref. [20]	99.6149%	33.5051%
	Ref. [22]	99.5951%	33.1216%
Cameraman	Ours	99.6231%	33.4488%
	Ref. [18]	99.5987%	33.2151%
	Ref. [19]	99.5739%	33.3015%
	Ref. [20]	99.6032%	33.5028%
	Ref. [22]	99.5897%	32.7981%
Peppers	Ours	99.5941%	33.5739 %
	Ref. [18]	99.4782%	33.4131%
	Ref. [19]	99.5801%	33.4466%
	Ref. [20]	99.6337%	33.5884%
	Ref. [22]	99.6111%	33.0147%

**Table 11 entropy-24-01827-t011:** Correlation coefficients.

Image	Directions	Ours	Ref. [18]	Ref. [19]	Ref. [20]	Ref. [22]
	Horizontal	0.0007	−0.0284	−0.0027	0.0039	0.0034
Baboon	Vertical	−0.0030	0.0147	0.0023	0.0103	−0.0019
	Diagonal	0.0080	0.0459	0.0088	−0.0070	0.0005
	Horizontal	−0.0020	0.0497	0.0047	−0.0063	0.0098
House	Vertical	−0.0147	0.0327	0.0030	0.0035	−0.0342
	Diagonal	0.0086	−0.0154	−0.0039	0.0103	0.0196
	Horizontal	−0.0055	0.0120	−0.0027	0.0047	0.0023
Cameraman	Vertical	−0.0008	0.0478	0.00025	0.0018	0.0044
	Diagonal	−0.0005	0.0354	0.0039	−0.0019	−0.0048
	Horizontal	−0.0029	−0.0654	−0.0008	0.0028	0.0051
Peppers	Vertical	0.0089	−0.0259	0.0083	−0.0017	−0.0049
	Diagonal	−0.0088	−0.0351	−0.0012	−0.0103	0.0078

**Table 12 entropy-24-01827-t012:** Comparison of ciphertext information entropy.

Methods	Baboon	House	Cameraman	Peppers
Ours	7.9974	7.9976	7.9972	7.9972
Ref. [18]	7.9865	7.9907	7.9716	7.9930
Ref. [19]	7.9609	7.9611	7.9581	7.9592
Ref. [20]	7.9974	7.9969	7.9974	7.9967
Ref. [22]	7.9672	7.9858	7.9773	7.9668

**Table 13 entropy-24-01827-t013:** Comparison of encryption algorithm runtimes (seconds).

Ref. [18]	Ref. [19]	Ref. [20]	Ref. [22]	Ours
1.84	1.69	1.91	0.83	1.12

## Data Availability

No new data were created or analyzed in this study. Data sharing is not applicable to this article.

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
