# Peer review of "An Image Encryption Algorithm Using Cascade Chaotic Map and S-Box"

_entropy, 2022, doi:10.3390/e24121827_

Round 1

Reviewer 1 Report

The authors proposed “An image encryption algorithm using cascade chaotic map and S-box. This article is well written, and the simulation results show its validity.  This article can be considered for publication after the incorporation of suggestions.

1.      What’s the uniqueness of your performance analysis? Maybe you need to discuss the advantages and disadvantages and compare the performance with other existing analyses.

2.      Is the proposed system resistant to Chosen-plaintext and Known-plaintext attacks.

3.      The author needs to give a proper situation for the applications.

4.      How to distribute the key should be considered and briefly discussed. 

Reviewer 2 Report

In this paper, a 2D cosine logistic sine map is proposed, and its application in an encryption algorithm is investigated. Cascade chaotic systems can help in key space, complexity, and sensitivity to initial conditions. The s-box method is proposed, and an encryption method based on the s-box is designed. Performance analysis of the encryption method shows good results. There are some comments as follows:

·         The authors should add a paragraph on the chaotic systems and their studies in the introduction. Also, they should consider some compression-encryption algorithms based on compressed sensing and chaotic oscillators. Encryption application based on various chaotic systems, like system crossing a cylinder, was also interesting.

·         There are some sentences in the paper which should be reconsidered. For example, “Chaos is a complex non-linear phenomenon that is irregular, random and yet contains certain laws.” What did the authors mean by random? What is their reference? Or they said, "In deterministic non-linear systems, chaos is stochastic and unpredictable in the long term.”

·         The authors should add some references for the cascade method.

·         I cannot understand how the authors said, "the more uniform the bifurcation diagram is respectively, the more stochastic the chaotic system is.” What is their measure?

·         Why is the resolution of information entropy much smaller than other measures?

·         The figure caption should contain enough information on the main results of showing the figure.

·         The authors should add more discussions comparing their results with the previous works. Only showing the results in the tables is not enough.

·         The conclusion of the paper should contain more discussions on the results of the paper.

Round 2

Reviewer 1 Report

The revised manuscript can be accepted.

Reviewer 2 Report

The revision can be accepted